# Dispersive Membrane Microextraction of Substituted Phenols from Honey Samples and a Brief Outlook on Its Sustainability Using Analytical Eco-Scale and Analytical GREEnness Metric Approach

**DOI:** 10.3390/membranes12070649

**Published:** 2022-06-24

**Authors:** Kumuthini Chandrasekaram, Yatimah Alias, Sharifah Mohamad

**Affiliations:** 1University Malaya Centre for Ionic Liquids, University of Malaya, Kuala Lumpur 50603, Malaysia; yatimah70@um.edu.my (Y.A.); sharifahm@um.edu.my (S.M.); 2Department of Chemistry, Faculty of Science, University of Malaya, Kuala Lumpur 50603, Malaysia

**Keywords:** dispersive membrane microextraction, mixed matrix membrane, honey, Analytical Eco-Scale, Analytical GREEnness Metric

## Abstract

Honey is part and parcel of our daily nutrition, but in recent times it has been reported to be tainted by the presence of polar substituted phenols purported from the use of pesticides, herbicides, antimicrobial agents, etc. Honey’s viscous nature and matrix complexity often result in analytical chemists resorting to derivatization for the detection of polar analytes such as substituted phenols. This study aims to overcome the matrix effect without derivatization and offer a more sustainable solution with notable sensitivity and selectivity using dispersive membrane microextraction alongside high-performance liquid chromatography (DMME–HPLC) with sporopollenin–methylimidazolium-based mixed matrix membrane (Sp–MIM-MMM). The DMME–HPLC approach successfully determined the presence of mono- and disubstituted phenols from unspiked honey samples with concentrations ranging from 7.8 to 154.7 ng/mL. The sustainability of the proposed method was also validated using the Analytical Eco-Scale (AES) and the Analytical GREEnness Metric (AGREE) where an excellent score of 94 and the encouraging score of 0.72 were recorded, respectively.

## 1. Introduction

Dispersive solid-phase microextraction (DSPME) is a non-exhaustive approach that helps minimize the setbacks encountered in the conventional exhaustive solid-phase extraction (SPE) method to bring about better selectivity, robustness, and versatility in the determination of analytes with high accuracy and precision in real samples [1,2,3,4]. DSPME can be defined as direct dispersion of solid sorbents into liquid samples for the extraction, isolation, and analyte clean-up procedures from complex matrices [2,3]. As the adsorbent is directly dispersed into the sample, closer contact is initiated for improved adsorption kinetics and method selectivity [5].

In DSPME, the adsorbent is fundamentally enhanced to provide better strength and stability which is practically achieved by chemically binding the adsorbent onto a stable core [1]. The binding serves both to strengthen the adsorbent and increase the reusability potential [1,2]. As such, DSPME is looked upon as a facile, rapid, cost-effective approach that shows good potential in the preconcentration of analytes with high extraction efficiency, high enrichment factor, and low sample volume consumption [1,3,4].

DSPME has been reported to be successful in the extraction and isolation of targeted analytes from complex matrices such as tetracyclines from milk samples [6], gold ions from jewelry production-generated wastewater [7], dyes from water samples [3,8], bis(2-ethylhexyl) phthalate from environmental water [9], drugs (naproxen and ibuprofen) from complex biological samples, etc. Nonetheless, DSPME has its fair share of challenges, too, as the adsorbent not only needs to selectively bind the targeted analytes, but also inherently retain them within their porous structure until subsequent desorption [5]. DSPME almost always requires the use of external energy sources such as mechanical agitation, ultrasonication, or vortexing, which plays a key role in minimizing time and the use of solvents [4,5,8]. Hence, the adsorbent has to be robust enough to handle the force applied while upholding its inherent adsorptive aptitudes [5].

Substituted phenolic compounds such as chlorophenols and nitrophenols form the basis of many industrial and chemical applications as either reagents or byproducts [10,11,12]. Chlorophenols are quite commonly used as disinfectants due to their anti-microbiological properties and are, therefore, present as an integral ingredient in most commercially available herbicides, insecticides, fungicides, as well as wood preservatives [13,14,15]. Nitrophenols, on the other hand, are widely used as a raw material or intermediate in various industries such as textiles, dyes, wood preservatives, pesticides, fungicides, herbicides, as well as pharmaceuticals, and thus are frequently detected in industrial effluents [16,17,18]. The US Environmental Protection Agency lists both chlorophenols and nitrophenols as priority pollutants and recommends a safety limit presence of lower than 30 ng/mL and 10 ng/L, respectively, in natural waters [17,19,20,21].

Recent reporting has indicated a worrying trend in the detection of these polar pollutants in everyday food items such as honey [12,13,14]. It is feared that due to the substituted phenols’ high resistance to biodegradation, they continue to retain their presence in food sources such as honey [12,22]. Researchers have propounded that the phenols could be sourced from the use of pesticides and herbicides in plants from which they are consequently transported by nectar-collecting traveling bees or via treated wooden beehives [13,14]. This study focuses on the use of a recently developed method [23], dispersive membrane microextraction (DMME), using sporopollenin–methylimidazolium-based mixed matrix membrane (Sp–MIM-MMM) alongside high-performance liquid chromatography (HPLC) to determine the presence of chloro- and nitro-substituted phenolsin honey samples procured from commercial and natural sources. DMME is an extension of DSPME where the Sp–MIM-MMM acts as a solid-phase adsorbent and has been reported to overcome the sample matrix effect quite successfully to determine the presence of the substituted phenols [22]. Following this, the sustainability of the DMME–HPLC method was also evaluated using the Analytical Eco-Scale (AES) [24] and the Analytical GREEnness Metric (AGREE) [25] approach.

## 2. Experimental Conditions

### 2.1. Reagent and Solution

The substituted phenol analytes, namely, 2,4-dinitrophenol (2,4-DNP), 2-nitrophenol (2NP), 4-nitrophenol (4NP), 2,4-dichlorophenol (2,4-DCP), 2-chlorophenol (2CP), and 4-chlorophenol (4CP) were purchased from Sigma Aldrich (St. Louis, MO, USA), while chromatographic-grade methanol was purchased from Merck (Steinheim, Germany) and used without further purification.

### 2.2. Instrumentation

Chromatographic analysis was conducted using high-performance liquid chromatography equipped with a diode array detector (HPLC-DAD) (Shimadzu, Japan). Accessories aiding the analysis included a pump (LC-20AT), a diode array detector (SPD-M20A), an autosampler (SIL-20A HT), a column oven (CTO-10AS VP), and a Shim-pack GIST C-18 reverse phase column (250 mm × 4.6 mm, particle size 5 µm) from Shimadzu, Kyoto, Japan. An isocratic mobile phase of 65% methanol: 35% ultrapure water (0.005% acetic acid) was used with a flow rate of 1 mL/min and a column temperature of 40 °C. The substituted phenols were detected with the wavelength of 280 nm, and analysis was completed in under 12 min.

### 2.3. Honey Samples

Honey samples were sourced from various readily available sources. In total, four different honey samples were used for the study, including a commercial brand readily available in any supermarkets in Malaysia (HS1), farm-harvested honey (HS2) from a healthy lifestyle-based store, honey fortified with weight-managing properties from an online portal (HS3), and the final sample was unadulterated natural honey (HS4) purchased from indigenous honey sellers of Cameron Highlands, Malaysia. The honey samples were prepared as described in the previously reported publication [23] with the linearity range of 50–500 ng/mL. The developed dispersive membrane microextraction alongside high-performance liquid chromatography (DMME–HPLC) method was then applied, and the presence of the substituted phenols was determined. All the samples were triplicated.

### 2.4. Dispersive Membrane Microextraction alongside High-Performance Liquid Chromatography

The DMME–HPLC determination of the substituted phenols including 2,4-dinitrophenol (2,4-DNP), 2-nitrophenol (2NP), 4-nitrophenol (4NP), 2,4-dichlorophenol (2,4-DCP), 2-chlorophenol (2CP), and 4-chlorophenol (4CP) was conducted following an optimized method described in the previous publication [23]. To a 5 mL solution of the honey sample, 3 units of Sp–MIM-MMM spheres (uniform diameter of 6.0 mm) were added, and the heterogeneous mixture was sonicated for 10 min, after which all the three Sp–MIM-MMM spheres were removed and briefly dried on lint-free tissue to avoid possible cross-contamination. The dried Sp–MIM-MMM spheres were then transferred into a vial containing 0.1 mL methanol and sonicated for 5 min. Subsequently, a methanol extractant containing the substituted phenol analytes was collected into HPLC vials for chromatographic analysis. All the sample runs were triplicated to ensure efficiency and accuracy. Figure 1 depicts the process in the form of a flowchart.

### 2.5. Sustainability Assessment Using the Analytical Eco-Scale and the Analytical GREEnness Metric (AGREE) Approach

A methodological evaluation of the sustainability of the proposed DMME–HPLC method was conducted using the Analytical Eco-Scale (AES) [24] method proposed by Gałuszka et al. (2012) as well as the Analytical GREEnness Metric (AGREE) and software [25] developed by Pena-Pereira et al. (2020).

## 3. Results and Discussion

### 3.1. Honey Samples

Table 1 outlines the validation of linearity, regression (R2), limit of detection (LOD), limit of quantification (LOQ), preconcentration factor (PF), matrix effect (ME), recovery, and relative standard deviation (RSD) for the honey samples. For the linearity range of 50–500 ng/mL, linear regression via a calibration plot was calculated to be 0.9976–0.9994. The preconcentration factor was calculated to be 111–168, whilst recovery and the RSD percentage were recorded within the accepted ranges of 88.18–100.34% and 0.28–8.16%, respectively. The honey samples recorded a significantly high matrix effect of 2.03–3.04%. Figure 2 depicts the detected chromatogram peaks of the substituted phenols in the honey samples.

Quantification of the detected peaks recorded concentration values between 7.8 and 154.7 ng/mL (Table 2). The analysis of the honey samples indicated the notable ability of an Sp–MIM-MMM to detect both low and high concentrations overcoming the matrix hurdle present in the honey samples (Figure 2). Out of the four samples analyzed, HS4 did not register any peaks, possibly because it was naturally sourced (hence, the honey was untainted by commercial preservative elements). HS1 and HS3 registered the presence of nitro-based phenols while HS2 registered the strong presence of chloro-based phenols. The overall findings of the study show that the adopted DMME–HPLC method exhibited both good selectivity and sensitivity towards the determination of the substituted phenols in the viscous honey samples.

Reported studies have indicated that chlorophenols are quite frequently detected in commercially produced honey, with bee pollination and the use of wooden beehives being the probable cause [13]. Chlorophenol is a widely used disinfectant for its anti-microbiological properties, and hence has been quite conspicuously used in herbicides, insecticides, fungicides, as well as wood preservatives [13,14]. The detection of chlorophenol in farm-harvested honey (HS2) does lend credence to the observation. Nitrophenols, on the other hand, are well-regarded as the infamous component of most commercially available pesticides and herbicides [18,26], and thus were detected in samples HS1 and HS3. The observation, therefore, supports the findings of Campillo et al. (2006) where the use of substituted phenols as disinfectants and pesticides resulted in their lingering presence in food items such as honey.

Additionally, it was observed that sample HS3 registered the presence of both disubstituted nitro- and chlorophenols, as well as of 4NP. Since the early 20th century, 2,4-DNP has been closely associated with weight loss regimes [27,28], and its presence has been reported in various dietary and detox products [28,29,30]. However, the weight loss-excitant drug has over the years been associated with numerous health hazards, including untimely death [28,29,30], and therefore, unsurprisingly, its usage in dietary products has been largely prohibited. Yet, time and again, the presence of 2,4-DNP has been inconspicuously integrated into purchasable products on the market as observed in HS3, which comes with a weight management tagline.

#### Influence of the Phenol Substituent Group and the pKa Value

As reported in a previous publication [31], phenol interaction towards an Sp–MIM-MMM is influenced by intermolecular interactions such as the π–π interaction between imidazolium and the aromatic ring of phenols as well as by potential hydrogen bonding due to the strong electronegative nitro-substituent group of phenols. Adhering to the reported work, the interaction trend of the selected phenols follows the order of 2,4-DNP > 4NP > 2NP > 2,4-DCP > 2CP > 4CP (Figure 3) in check to their respective pKa. It is observed that 2,4-DNP with a low pKa value is the most favored while the 4CP with a high pKa value is the least favored. In general, the interaction strength decreases with an increase in the pKa value. This occurrence propounds the importance of ionic dissociation of the phenols in the extraction process. Most phenols act like weak acids in aqueous solutions wherein apart from solution pH, their respective physicochemical properties play an important role. Therefore, with the lowest pKa value, the effectual ionic dissociation of 2,4-DNP at mildly acidic conditions promotes the strongest interaction [32,33].

It is also notable that nitro-substituted phenols recorded stronger interaction in both mono- and disubstituted configurations. This phenomenon could be due to a nitro substituent which possesses two oxygen atoms, exhibiting stronger electronegativity and inciting hydrogen bonding interactions [32,34]. The presence of two substituent groups also increases the interaction potential [35,36], with both 2,4-DNP and 2,4-DCP recording an increased peak area in comparison to their monosubstituted counterparts.

### 3.2. Green Analytical Chemistry Metric-Based DMME–HPLC Sustainability Assessment

Green analytical chemistry (GAC) is a branch of the humongous tree of green chemistry where researchers fight a daily battle to bring sustainable solutions to everyday problems. GAC could be briefly described to represent the research and development of miniaturizing analytical procedures by limiting the use of hazardous substances, minimizing the analysis stratagem, encouraging energy efficiency, and inhibiting secondary waste generation by promoting the use of natural biomass [24,37]. In a nutshell, GAC aims to support the future of our environment and health via sustainable research and development. Much influenced by GAC’s cause, we implemented it in our developed DMME–HPLC method. Primarily, the mixed matrix membrane (Sp–MIM-MMM) consists of biodegradable plant-based sporopollenin biopolymer and cellulose triacetate, which impedes secondary waste generation.

On the analytical front, due to the strong interaction between the substituted phenols and the Sp–MIM-MMM, a simple sonication method enables dynamic adsorption with less energy usage. The subsequent elution, too, is easily achievable using sonication despite the use of methanol which has been reduced to the bare minimum. Phenols are polar molecules, hence polar solvents such as methanol are generally more efficient in desorbing phenols from adsorbents [38]. As can be observed from Table 3, methanol is a preferred solvent in most solid-phase extractions of the substituted phenols.

Apart from that, the DMME–HPLC method proposed essentially minimizes the analysis stratagem to two steps of sonication followed by HPLC analysis. Table 3 compares the reported works on solid-phase extraction of substituted phenols from honey samples. Solid-phase extraction (SPE) from honey-based samples usually faces a strong adversary in the sample matrix. As such, most reported SPE methods require preparatory steps such as headspace modification, guard column enhancement, and use of derivatives. Providently, in this study, with strong intermolecular interactions incited between the Sp–MIM-MMM and the substituted phenols, additional preparatory steps could be averted.

#### 3.2.1. Analytical Eco-Scale

Analytical Eco-Scale (AES) is a form of the green analytical metric approach whereby the sustainability or greenness of a chosen method is fitted into a model system and evaluated based on its environmental reverberations with consideration of the type of reagents, minimal use of solvents, nominal energy consumption, abstaining from hazards, and eradicating waste generation [37,44]. To evaluate the method-oriented sustainability metrics of the proposed DMME–HPLC method, the analytical eco-scale model developed by Galuszka et al. (2012) was adopted [24,44]. The model is based upon a point system whereby penalty points are allotted to the type of reagents, energy consumption, hazards posed, and waste generated. Subsequently, the cumulative penalty points are subtracted from the total value denoted to be 100 points, the ideal value [24]. Galuszka et al. (2012) proposed the categorization of the analytical eco-scale into four main segments with 100 being ideal, above 75 being excellent, 74–50 being acceptable, and anything below 50 being inadequate. Table 4 summarizes the penalty point calculations for the proposed DMME–HPLC method with reference to the AES [24,45]. By virtue of this, the AES value of our method fell under the excellent category with a score of 94.

#### 3.2.2. Analytical GREEnness Metric (AGREE)

Where the AES approach uses penalty points to assess the sustainability of an analytical method, the Analytical GREEnness Metric (AGREE) delves into the 12 SIGNIFICANCE GAC to comprehensively evaluate greenness [25,46]. Thanks to the joint efforts of green chemistry researchers from Universidade de Vigo, Spain, and Gdansk University of Technology, Poland, an easily adaptable Analytical GREEnness calculator v.05beta is available for free access [25]. The software enables fellow aspiring green analytical chemists to better evaluate respective research sustainability and promote greenness. Figure 4 displays the generated Analytical GREEnness Metric (AGREE) for the proposed DMME–HPLC method.

The circular representation takes on the hue of dark green being the ideal sustainable/green method with a point score of 1.0, and red being the least ideal method with the lowest score of 0.0. The nearer the central core score is to the ideal 1.0 value, the greener the adopted method of study. Our study scored a 0.72 value denoting its applicable sustainable approach, albeit with a few improvements. The red-hued score for reagents (No. 10) was based on the consideration the solvent was not from bio-based sources which offer better sustainability. Therefore, it is possible to improve the greenness of our method by sourcing our methanol from bio-based industries such as biomass feedstocks [47]. The method (No. 5) registered a yellow hue which falls under the acceptable category, mainly due to the method being conducted manually; however, with the optimized parameters, future automation is likely to improve the score. Orange hues (Nos. 1 and 3) were registered for the sampling procedure and the analytical device positioning as the sample was pretreated prior to injection for HPLC. The choice of the instrument (No. 9) registered a yellow hue for the use of HPLC. However, it is to be noted that the instrument run time per sample was at 12 min, and the system had energy consumption of <1.5 kWh [24].

Generally, both AES and AGREE propound that the proposed DMME–HPLC method offers a potentially satisfying sustainable approach. The method could indeed be improved further by including automation, bio-sourced reagents, and potential in-line or on-line positioning in the upcoming future.

## 4. Conclusions

The study exhibited favorable outcomes in overcoming the matrix hurdle presented by viscous honey-based samples and in providing a sustainable analytical method toward a better adoptable greenness perspective. The dispersive membrane microextraction alongside high-performance liquid chromatography (DMME–HPLC) was able to successfully discern the highly polar substituted phenols without the commonly adopted derivation step while still exhibiting good sensitivity and selectivity. The DMME–HPLC method also recorded commendable performance on the Analytical Eco-Scale (AES) and in the Analytical GREEnness Metric (AGREE), with the scores of 94 and 0.72, respectively. Therefore, the proposed DMME–HPLC method stimulates yet another sustainable approach for analytical chemists to start a green future.

## Figures and Tables

**Figure 1 membranes-12-00649-f001:**
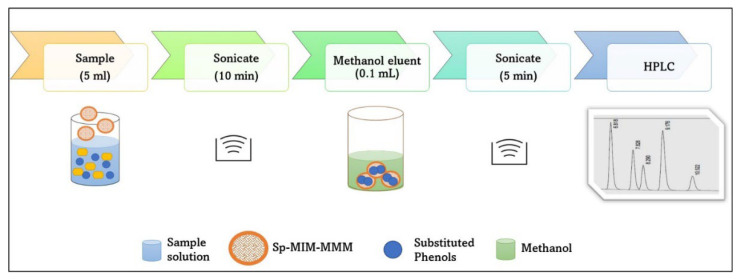
Dispersive membrane microextraction (DMME) of the substituted phenol analytes from the honey samples.

**Figure 2 membranes-12-00649-f002:**
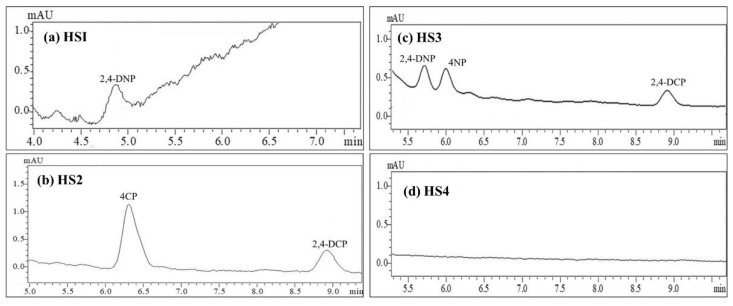
Chromatogram of the substituted phenols detected in the unspiked honey samples (HS1–4) using the DMME–HPLC method.

**Figure 3 membranes-12-00649-f003:**
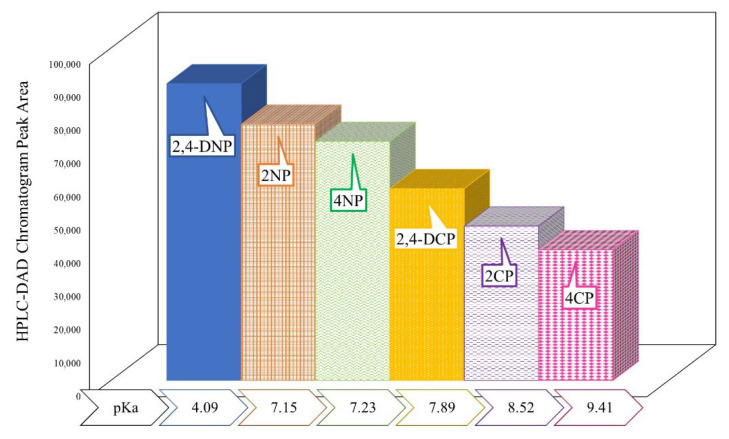
Interaction trend and pKa values of the substituted phenols.

**Figure 4 membranes-12-00649-f004:**
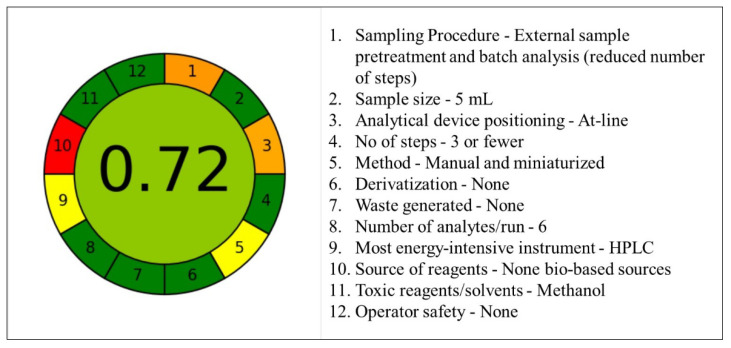
Analytical GREEnness Metric (AGREE) for the proposed DMME–HPLC method.

**Table 1 membranes-12-00649-t001:** Method validation for the substituted phenols analytes detected in the honey samples.

Method Validation Parameters	Substituted Phenol Analytes					
	2,4-DNP	4NP	2CP	4CP	2NP	2,4-DCP
Linearity (ng/mL)	50–500	50–500	50–500	50–500	50–500	50–500
R^2^	0.9994	0.9982	0.9976	0.9977	0.9976	0.9994
LOD (ng/mL)	2.93	5.03	7.39	5.77	5.8	2.93
LOQ (ng/mL)	7.34	12.57	18.49	14.43	14.49	7.34
Preconcentration factor	135	121	168	121	111	135
Matrix effect (%)	2.03	2.70	3.04	2.33	2.22	2.03
	Spiked (50 ng/mL)					
Recovery (%)	100.29	97.30	99.90	95.90	88.18	100.29
RSD (%)	5.76	3.29	5.70	8.16	6.92	5.76
	Spiked (100 ng/mL)					
Recovery (%)	100.34	99.02	96.04	97.45	91.35	100.34
RSD (%)	0.28	0.68	0.41	4.51	6.83	0.28

**Table 2 membranes-12-00649-t002:** The concentrations of the substituted phenols detected in the honey samples.

Honey Samples	Quantitative Concentration Detected (ng/mL)					
	2,4-DNP	4NP	2CP	4CP	2NP	2,4-DCP
HS1	7.8	*N.D.*	*N.D.*	*N.D.*	*N.D.*	*N.D.*
HS2	*N.D.*	*N.D.*	*N.D.*	154.7	*N.D.*	72.7
HS3	34.4	33.9	*N.D.*	*N.D.*	*N.D.*	17.4
HS4	*N.D.*	*N.D.*	*N.D.*	*N.D.*	*N.D.*	*N.D.*

*N.D.* = not detected.

**Table 3 membranes-12-00649-t003:** Reported works on solid-phase extraction of substituted phenols from honey samples.

No	Method	Reagents	Instrument	Reference
1	Solid phase microextraction (SPME) with in-situ derivatization	Sodium chloride; potassium carbonate; acetic anhydride	Homogenizer; GC	[13]
2	Magnetic three-dimensional graphene solid-phase extraction	Hydrochloric acid; alkaline methanol	Vortex; HPLC	[39]
3	Dispersive micro-solid-phase extraction (DMSPE) combined with headspace solid-phase micro-extraction (HS-SPME)	Ethanol; sodium chloride	Centrifuge; hotplate; GC	[40]
4	Zn/Co bimetallic metal–organicframework for magnetic solid-phase extraction	Alkaline methanol; hydrochloric acid	Shaker; centrifuge; HPLC	[41]
5	Phenylboronic acid-based hyper-crosslinked polymers solid-phase extraction	Methanol; sodium hydroxide; hydrochloric acid	HPLC-DAD	[42]
6	Imine-linked covalent organic framework for solid-phase extraction	Methanol; acetonitrile	Vacuum pump; HPLC	[43]
7	Mixed matrix membrane-based dispersive membrane microextraction (DMME)	Methanol	Sonicator; HPLC-DAD	This study

**Table 4 membranes-12-00649-t004:** Penalty point calculations for the proposed DMME–HPLC method.

Parameter	Amount Penalty Points (PPs)	Hazard Penalty Points (PPs)	Reagent Penalty Points (Amount PPs × Hazard PPs)
Reagent Methanol	0.1 mL (1)	Flammable: danger, Category 2 (2) Toxic: warning, Category 3 (1) Health hazard: danger, Category 1 (2)	5
Instrument	Energy consumption (PPs)		Energy penalty points
Sonicator HPLC	<0.1 kWh per sample (0) <1.5 kWh per sample (1)		1
Total penalty points			6

## Data Availability

Not applicable.

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
