# Peer review of "Dispersive Membrane Microextraction of Substituted Phenols from Honey Samples and a Brief Outlook on Its Sustainability Using Analytical Eco-Scale and Analytical GREEnness Metric Approach"

_membranes, 2022, doi:10.3390/membranes12070649_

Round 1

Reviewer 1 Report

In this interesting paper, the authors investigated the possibilities to overcome the matrix effect of honey samples for the determination of substituted phenols, and offer a more sustainable solution with notable sensitivity and selectivity with dispersive membrane microextraction coupled with high performance liquid chromatography (DMME-HPLC) using sporopollenin-methylimidazolium based mixed matrix membrane (Sp-MIM-MMM). The sustainability of the proposed method was also validated using Analytical Eco Scale and Analytical GREEnness Metric.

The research was done and the manuscript was written in a professional manner. The results are discussed in detail and competently explained.

I do not have the comments and suggestions (changes). I suggest that this manuscript should be accepted without changes, but with small professional language corrections.

Author Response

The author thanks esteemed Reviewer 1 for their time and encouragement.

Our profound apologies on the unintended language errors.

The manuscript has since been proofread by a professional and amended.

Reviewer 2 Report

This study aims to detect the phenolic pollutants in honey using extraction and chromatographic method, followed by greenness assessment of proposed method. This is interesting idea.

Language: must be checked by expert, several language mistakes are there in whole text.

Title: Agree should be AGREE as this is abbreviation of the method’s name

Line 68: developed by us, please try to use passive expressions in whole manuscript instead of active ones i.e. avoid our, we , us…etc

Line 90: C-18 re-89 verse phase column (250 × 4.6 mm, particle size 5 μm), you mean 250 mm?

Line 92: 40 oC, degree sign should be superscript

110: described in our prior publication., which reference?

Regarding greenness assessment:

-        Mobile phase and extraction procedure include methanol in high amount, which is well know to be toxic , hence do you think the high greenness score you got is realistic?

-        A recent work focused on AGREE tool application and importance of considering the weights on the 12 SIGINIFICANCE criteria: https://www.mdpi.com/2297-8739/9/6/147 , did you consider weights when adjusting the software while using AGREE software?

-         

Author Response

The author thanks esteemed Reviewer 2 for their time and observations.

Our profound apologies on the unintended errors.

The manuscript has been proofread by a professional and the errors amended.

On the greenness assessment,

With deep respect, the current study is an early attempt for us to understand our adsorbent and its potential sustainability hence we have focused on the performance of our Sp-MIM-MMM. We have also taken cue from Tobiszewski, M. (2016) where he has in his study categorized methanol under the green scheme. Methanol is observed to be still generally less toxic among the most commonly used solvents and additionally the sourcing of methanol from lignocellulosic biomass sources could propel its greenness further. We do believe and have observed in our study on the potential to further improve and gain better greenness with our method.

Our apologies as for the current study we did not attempt weight adjustment. Our objective here was to explore on the initial possibilities of sustainability via the use of AES and AGREE approach. We do have plans on improving and enhancing our study further with sustainable approaches and extend the exploration on the use of greenness evaluating metrics.